# Genome-Wide Association Analysis Reveals Novel Loci Related with Visual Score Traits in Nellore Cattle Raised in Pasture–Based Systems

**DOI:** 10.3390/ani12243526

**Published:** 2022-12-13

**Authors:** Pamela C. Machado, Luiz F. Brito, Rafaela Martins, Luis Fernando B. Pinto, Marcio R. Silva, Victor B. Pedrosa

**Affiliations:** 1Department of Animal Sciences, State University of Ponta Grossa, Ponta Grossa 84030-900, PR, Brazil; 2Department of Animal Sciences, Purdue University, West Lafayette, IN 47907, USA; 3Department of Animal Science, Federal University of Bahia, Av. Adhemar de Barros 500, Ondina, Salvador 40170-110, BA, Brazil; 4Melhore Animal and Katayama Agropecuaria Lda, Guararapes 16700-000, SP, Brazil

**Keywords:** beef cattle, weight performance, SNP effect, tropical cattle, genomic regions

## Abstract

**Simple Summary:**

Genome-wide association studies attempt to understand the genetic structure through the expression of genes that influence a given productive trait. This work studied the associations between single nucleotide polymorphism with visual score traits in beef cattle, identifying genomic regions and candidate genes with greater effects for body conformation, precocity, and muscularity. Several novel genomic regions that had not been previously identified in the literature were revealed. Our results will contribute to a better understanding of the molecular mechanism involved in Zebu cattle growth, serving as a base reference in adjacent research related to the application of molecular markers in the detection of new genomic regions related to traits of economic interest in beef cattle.

**Abstract:**

Body conformation traits assessed based on visual scores are widely used in Zebu cattle breeding programs. The aim of this study was to identify genomic regions and biological pathways associated with body conformation (CONF), finishing precocity (PREC), and muscling (MUSC) in Nellore cattle. The measurements based on visual scores were collected in 20,807 animals raised in pasture-based systems in Brazil. In addition, 2775 animals were genotyped using a 35 K SNP chip, which contained 31,737 single nucleotide polymorphisms after quality control. Single-step GWAS was performed using the BLUPF90 software while candidate genes were identified based on the Ensembl Genes 69. PANTHER and REVIGO platforms were used to identify key biological pathways and STRING to create gene networks. Novel candidate genes were revealed associated with CONF, including *ALDH9A1*, *RXRG*, *RAB2A*, and *CYP7A1*, involved in lipid metabolism. The genes associated with PREC were *ELOVL5*, *PID1*, *DNER*, *TRIP12*, and *PLCB4*, which are related to the synthesis of long-chain fatty acids, lipid metabolism, and muscle differentiation. For MUSC, the most important genes associated with muscle development were *SEMA6A*, *TIAM2*, *UNC5A*, and *UIMC1*. The polymorphisms identified in this study can be incorporated in commercial genotyping panels to improve the accuracy of genomic evaluations for visual scores in beef cattle.

## 1. Introduction

Nellore cattle (*Bos taurus indicus*) is one of the main beef cattle breeds raised in tropical regions, especially in South America. These animals are raised mostly in pasture-based systems and harsh environmental and geographical conditions. In general, animals that exhibit superior body attributes, such as better conformation and muscling, tend to be more profitable [1]. The characteristics evaluated based on easily measured visual scores can be used as productive efficiency indicators and selective breeding goals. Visual score traits related to carcass finishing and body development in beef cattle have moderate heritabilities [2,3]. Thus, selecting animals with superior visual scores improves growth rate, muscling, and fat deposition. Because of that, incorporating visual evaluation score traits in breeding programs is feasible and can be highly recommended [1]. In addition, these traits show favorable genetic correlation with other economically important traits such as body weight at different ages [4,5], carcass finishing [6,7], and carcass yield [8,9]. Therefore, these traits may be considered as selection criteria in Nellore cattle breeding programs for improving beef production. Additionally, visual score traits are usually evaluated at early ages, before the final finishing phase, facilitating earlier selection of breeding bulls [10]. The measurement of carcass traits in slaughter plants can be more expensive and time-consuming than visual scoring, making it more difficulty to genetically select for objective traits, especially in developing countries where there are less research and development investments in the agricultural sector. Genetic selection for slaughter-plant measurements is also more challenging for breeding candidates, as they would not have own performance prior to slaughter [11]. This constrain can be solved with genomic selection.

Genomic data are widely used to improve the reliability of genetic merit estimates in livestock species [12], especially for traits that are difficult or expensive to measure such as carcass and meat quality traits. The identification of candidate genes or genomic regions with significant effects on traits related to meat production and carcass quality can aid in the design of genomic selection strategies and in the development of genomic tools (e.g., genotyping arrays). Genome-wide association studies (GWAS) enable the identification of single nucleotide polymorphisms (SNPs) associated with genes of substantial effect on the expression of traits related to meat quality, providing a list of candidate genes for genome mapping [13,14]. In recent years, GWAS have been widely performed for livestock species, especially in cattle populations [15], enabling the identification of many markers and genomic regions related to meat quality and carcass traits in different beef cattle breeds [16,17], as complied in the Animal QTL database, with 22,663 QTLs associated with meat and carcass traits. However, studies with visual score traits related to carcass in beef cattle populations are scarce, particularly when considering Zebu (*Bos taurus indicus*) populations or cattle raised in challenging production systems such as pastures in tropical regions.

Multiple genomic windows were identified accounting for small percentages of the total additive genetic variation for visual score traits in Nellore cattle, indicating the polygenic nature of these traits [18]. SNPs associations were also identified with for conformation and precocity traits in Nellore cattle, which were linked to genes involved in body growth and development, energy and protein metabolism, and homeostasis [19]. In cattle, several authors identified genes related to skeletal muscle growth and development [20,21], as well as genes related to muscle mass increase [22], indicating that these traits are genetically controlled. Thus, the genetic background of visual score traits linked to muscle growth and precocity in pasture-raised beef cattle needs to be further explored. Furthermore, knowledge of metabolic pathways and gene co-expression networks allows a better understanding of specific biological mechanisms underlying the phenotypic variability in the traits of interest [23]. Therefore, the aim of the present study was to perform a single-step GWAS (ssGWAS) to identify genomic regions, candidate genes, and their biological functions associated with conformation, precocity, and muscling scores in Nellore cattle raised in pasture-based systems.

## 2. Materials and Methods

### 2.1. Phenotypes, Genotypes, and Pedigree

This study used visual score data from in vivo evaluation of conformation, precocity, and muscling measured at 18 months of age in 20,807 Nellore cattle born between 2009 and 2018. A single trained technician performed the visual evaluation for CONF, PREC, and MUSC, first observing the whole management group, divided by males and females, evaluating the average animal frame, which is considered as a baseline for that particular group of individuals. Then, scores from 1 (lower characterization) to 5 (higher characterization) were attributed to the animals [24]. The CONF is a visual score that estimates the body area seen from the side, evaluating body length and rib depth, in which greater areas are associated with higher scores. For PREC, higher scores are assigned to animals with superior rib depth in relation to the length of their limbs, in which superior fat coverage indicates animals with higher score values. MUSC is evaluated by visual evidence of muscle volume. Higher scores are assigned to robust animals with higher amounts of convex muscles.

The 20,807 animals were progeny of 10,088 cows and 416 bulls, raised in the herds of Katayama Ltd. a farms located in the Brazilian states of São Paulo, Mato Grosso, and Mato Grosso do Sul. The animals were maintained on high-quality pasture system containing 35% *Brachiaria brizantha* and 65% *Panicum maximum*, receiving salt and mineral supplementation. The complete pedigree file contained 39,503 animals comprising at least three generations. A total of 2775 animals were genotyped with the 35 K GGP-Indicus panel (Neogen Company, Lansing, MI, USA) containing 35,247 SNPs. The genetic material was extracted from hair follicle samples of the animals using a phenol-chloroform extraction protocol [25]. The concentration (ng/µL) and purity of the DNA were determined in a Nanodrop spectrophotometer (Thermo Fisher Scientific, Waltham, MA, USA).

### 2.2. Data Quality Control

Quality control was performed using the BLUPF90 package [26]. As quality control criteria, animals and SNPs with a call rate < 0.90, non-autosomal SNPs, SNPs with unknown position, duplicated SNPs, SNPs with a minor allele frequency (MAF) < 0.05, SNPs with extreme deviation from Hardy–Weinberg equilibrium (*p* ≤ 10^−5^), and strongly correlated SNPs (r^2^ > 0.90) were removed. After quality control, the final set for the association analyses consisted of 31,737 SNPs. Finally, phenotypic records that exceeded three standard deviations from the mean within the contemporary group were considered outliers and removed from the database.

### 2.3. Single-Step Genome-Wide Association Study (ssGWAS)

The ssGWAS method was used for the association analyses employing the BLUPF90 family of programs [26]. The AIREMLF90 module [27] was used to estimate variance components and genetic parameters considering a convergence criterion of 10^−12^. The PREGSF90 module [28] was applied to create the hybrid relationship matrix [29], which includes genotyped and non-genotyped animals. Finally, the postGSf90 module was used to back solve the genomic estimated breeding values (GEBV) and obtain the solutions of SNP effects for each trait.

The three traits were analyzed using the animal model, as follows:**y** = **Xb** + **Za** + **e**,
where **y** is the vector of phenotypic observations; **X** is the incidence matrix relating the phenotypes to the fixed effects; **b** is the vector of fixed effects, which include age at measurement as linear and quadratic covariates and the contemporary group (farm, year/season of birth, management group, and sex); **Z** is the incidence matrix relating the random effects to the phenotypic records; **a** is the vector of additive genetic effects; and, **e** is the vector of residual effects.

The variances of **a** and **e** can be written as:$$\mathrm{Var}=\left[\begin{array}{cc} H\sigma^{2}{}_{a} & 0\\ 0 & I\sigma^{2}{}_{e} \end{array}\right],$$
where **σ^2^_a_** is the direct additive genetic variance and **σ^2^_e_** is the residual variance. **H** is the relationship matrix combining the pedigree and genomic relationship matrices as described by and **I** is an identity matrix [29]. The inverse of the **H** matric is represented by the equation:$$H^{-1}=A^{-1}+\left[\begin{array}{cc} 0 & 0\\ 0 & G^{-1}-A_{\;22}^{-1} \end{array}\right],$$
where **A** is the pedigree relationship matrix for all animals; **A₂₂** is the relationship matrix of genotyped animals, and **G** is the genomic relationship matrix [30].
$$G={\mathrm{WDW}}^{\prime }\frac{1}{q}$$
where **W** is a matrix relating genotypes of each locus to each individual; **D** is a matrix of weights attributed to the SNPs (initially **D** = **I**), and **q =** $\overset{M}{\underset{i=1}{\sum }}{2p}_{i}{(1\;-\;p}_{i})$ is a normalization factor.

### 2.4. Estimation of SNP Effects

The effects of SNPs and their weights were estimated based on three iterations [31].

D(t) = **I**

$$G_{\left(t\right)}=\frac{{\mathrm{WD}}_{(t)\;}W^{\prime }}{\sum_{i=1}^{M}{2p}_{i}{(1\;-\;p}_{i})}$$
where **t** is the iteration number. The SNP effects (ȗ) were obtained as:**ȗ** = λ**DW′G** − 1**ȃ**_g_ = **DW′[WDW′]** − 1 **ȃ**_g_,
where **ȃ**_g_ is a vector of genetic effects of genotyped animals, which is represented as the function of SNP effects (**ȃ_g_** = **Wu**); **W** is the matrix of genotypes for each locus; **û** is the vector of SNP effects; **λ** is the variance ratio [30]; **D** is the matrix of weights attributed to the SNPs, and **G** is the genomic relationship matrix obtained as described above. The following model was used to calculate the SNP weights:$$d_{i(t=1)}={ȗ}_{i(t)}^{2}{2p}_{i}{(1\;-\;p}_{i}),$$
where **i** = SNP i. Finally, the program calculates **G** with the new weights attributed to the markers:$$G_{\left(t+1\right)}=\frac{{\mathrm{WD}}_{(t+1)\;}W^{\prime }}{\sum_{i=1}^{M}{2p}_{i}{(1-\;p}_{i})}.$$

The results of ssGWAS are presented as the proportion of total additive variance explained by a genomic window of 10 adjacent SNPs [32]:$$\frac{{\mathrm{var}(\mathrm{window}}_{i})}{\alpha_{a}^{2}}\times \;100\%\;=\frac{\sum_{j=1}^{10}{\mathrm{var}(ȗ}_{j})}{\alpha_{a}^{2}}\times \;100\%,$$
where window_i_ is the additive breeding value of genomic window i; **σ^2^_a_** is the total additive genetic variance for the trait, and **û_j_** is the effect of SNP i within genomic window i.

### 2.5. Identification of Candidate Genes and Functional Analyses

The chromosome regions containing the most relevant SNPs were explored through the identification of genes in genomic windows that explain more than 1.0% of the total additive genetic variation. These windows were selected as possible QTL associated with the three visual traits in Nellore cattle. The list of candidate genes was generated with the BioMart tool considering 120-kb intervals (threshold defined based on the extent of linkage disequilibrium in the studied population) using the Ensembl Genes database and the bovine reference genome assembly ARS-UCD1.2.

To improve our understanding of the biological mechanisms and processes shared by the candidate genes, gene ontology (GO) term enrichment analysis was performed using the functional annotation tools of the PANTHER database [33]. The REVIGO software was used to illustrate ontological coverage of the genes in blocks [34]. Gene networks were generated using the STRING platform [35] to identify and facilitate the understanding of interactions between genes and their contribution to the expression of complex and polygenic production traits in beef cattle.

## 3. Results

### 3.1. Descriptive Statistics and Variance Components

Table 1 shows the descriptive statistics, variance components, and heritabilities for the visual score traits measured at yearling in Nellore cattle. The number of individuals and the minimum and maximum scores were the same for all traits since they are weighted by the same methodology at subsequent times. The heritabilities estimated for the three traits were of moderate magnitude, ranging from 0.33 for conformation to 0.38 for muscling. The genetic correlation (±S.E.) between CONF and PREC was 0.53 ± 0.02, CONF and MUSC was 0.52 ± 0.01, and PREC and MUSC was 0.58 ± 0.01.

The frequency of scores 1, 2, 3, 4, and 5 were 7%, 17%, 29%, 26%, and 21%; 6%, 18%, 28%, 26%, and 22%; and 7%, 19%, 29%, 26%, and 19%, for CONF, PREC and MUSC, respectively. The variability observed indicate that there were animals with low subcutaneous fat coverage and low muscle mass and other with excellent carcass finishing and muscle development at specific body areas such as forearm, shoulder, loin, and rump.

### 3.2. Single-Step GWAS

The results of ssGWAS are presented as the proportion of the total additive genetic variance explained by windows of 10 adjacent SNPs. For conformation, 27 genomic regions explaining more than 1.0% of the total additive genetic variance for the trait were identified on chromosomes Chr3, Chr5, Chr14, Chr20, Chr21, Chr27, and Chr29 (Figure 1), as presented in Table 2.

For this trait, the genomic region with the highest peak was found on Chr14 26,217,826:26,253,265, linked with the *RAB2A* gene. For PREC, 26 genomic regions were found on Chr2, Chr3, Chr8, and Chr23 (Table 3). These genomic regions harbor 26 genes and individually explained from 1.04% to 5.28% of the total additive genetic variance. Among the genomic regions associated with PREC, the highest peak was found on Chr23 25,165,091:25,171,927, which contains the *GSTA5* gene. For MUSC, 19 genomic regions were identified across Chr1, Chr7, Chr9, Chr16, and Chr21 (Table 4). These regions harbored 19 genes, with *SEMA6A* located in the highest Manhattan peak (Figure 1). In addition to the location and chromosome position of the genes identified by the markers for the three traits, Table 2 also provides information about the proportion of the total additive genetic variance explained by each region, a parameter that is important to explain the influence of each gene on the trait analyzed.

### 3.3. Functional Analyses and Gene Networks

Analysis of the relevance of ontology terms revealed that 11 of the 27 candidate genes associated with visual conformation scores had known GO functions. Each of these genes was found to be involved in multiple biological processes related to cell differentiation and adhesion, protein synthesis and modification, macromolecule deacylation, anatomical structure development, and lipid homeostasis (Appendix A). The *AEBP2* gene was present in 14 different GO terms. The ontological coverage of genes involved in the expression of conformation is illustrated in blocks in Figure 2, where each block represents one GO term, and its size is proportional to the number of genes involved. The largest block shown in yellow involves multiple processes linked to cell junction organization, with important genes such as *APBE2*, *UBXN2B*, and *CTNND2*.

Analysis of PREC revealed that eight of the 26 genes identified were linked to 29 different GO terms related to biological, metabolic, and molecular processes (Appendix A). *ELOVL5* was the gene most frequently involved in different processes related to lipid metabolism. Figure 3 shows the ontological coverage of candidate genes involved in the phenotypic expression of PREC in blocks, in which events that regulate the mitochondrial metabolism of reactive oxygen species, response pathways to xenobiotic stimuli, and the metabolism of long-chain fatty acids are the main processes related to the candidate genes identified.

For MUSC, GO analysis revealed eight multifunctional genes involved in diverse biological and metabolic processes, totaling 102 GO terms (Appendix A). The *SEMA6A* gene is involved in 41 biological processes, with those linked to nervous system development being the most known functions. Regarding ontological coverage of the genes involved in the phenotypic expression of MUSC (Figure 4), the largest block shown in red involved multiple processes related to RNA synthesis and modification, highlighting the *UIMC1* and *SCAF8* genes. The remaining genes are associated with energy metabolism, growth, response to stimulus, organic compounds, glucose homeostasis, and axonogenesis.

Figure 5 shows the interaction between candidate genes in the expression of the three traits (conformation, precocity, and muscling) using the STRING database. Although some genes do not show an interaction with each other and remain separate, analysis of the co-expression network identified 36 pairs of significantly correlated genes, involving 41 genes in total. The connection among genes is represented by the lines linking one gene to the other, with an intense and distinct staining according to the strength of the interaction. This fact supports the argument that some genes interact towards the phenotypic expression of the traits evaluated.

## 4. Discussion

The heritability estimates for conformation (0.33), precocity (0.37), and muscling (0.38) were of moderate magnitude. Studies investigating these traits in beef cattle have also reported moderate to high heritabilities (0.23 to 0.44) [36,37,38]. These results confirm the hereditary and selective potential of PREC, CONF, and MUSC based on visual scores, with potentially high genetic responses due to direct selection.

The ssGWAS method identified different SNP markers spread across 29 autosomal chromosomes. Because the animals under study were raised in pasture systems, it is expected that new genomic regions related to potential candidate genes have been revealed. In GWAS studies, it is important that populations reared in different systems are investigated in order to cover all possible genes that are responsible for determining relevant traits in livestock species.

Many markers were located next to genes that may directly act on the expression of the traits studied. However, there are no previous reports of the effect of some genes on carcass traits in cattle, a fact that justifies the need to expand our knowledge about the actions of these genes both in cattle and in other livestock species. For CONF, genomic regions were found close to six genes: *TMCO1*, *UCK2*, *ALDH9A1*, *MGST3*, *LRRCR2*, and *RXRG*. Under the influence of the *TMCO1* gene, this region may be related to calcium ion homeostasis in endoplasmic reticulum membranes and to energy homeostasis [39], as previously reported for commercial pigs [40]. In cattle, an association with *TMCO1* was demonstrated related with the efficiency of the humoral immune response to infection with *Ostertagia ostertagi*, a gastrointestinal nematode with a high incidence in tropical herds [41].

The *ALDH9A1* gene (aldehyde dehydrogenase 9, family member A1) is considered to be involved in the degradation of fatty acids since it is considered a precursor of carnitine [42] a metabolite that aids in fat burning through mitochondrial oxidation of long-chain fatty acids [43]. Thus, the endogenous biosynthesis of carnitine is one of the mechanisms of fat deposition in the carcass of beef cattle [44]. The *ALDH9A1* gene functions linked to energy metabolism, including oxidative phosphorylation, pyruvate metabolism, and gluconeogenesis [45], provide substances that are source of energy supply for the maintenance of the animal’s basal metabolism.

Also located on Chr3, previously report linked the importance of the *RXRG* gene for conformation in analyses of growth traits in Chinese indigenous cattle [46]. The influence of *RXRG* on CONF can be explained by the activity of gamma receptors that regulate genes involved in adipocyte transcription factor activation, mediating lipid and glucose homeostasis [47]. This gene was also linked to pathways related to response to acid chemical (GO:0001101) and body development, including anatomical structural development (GO:0048856), developmental process (GO:0032502), and cell differentiation (GO:0030154). Alterations in *RXRG* expression promote changes in liver transcription levels related to fat metabolism [48].

On Chr5, the most relevant genes for CONF were *PLEKHA5* and *AEPB2.* Due to its pleckstrin homology (PH) domain, the *PLEKHA5* gene acts on cell-cell interaction, in addition to guiding proteins to the cell membrane. This role in intracellular signaling has been previoulsy reported to explain the influence of this gene on milk fat content in dairy cattle [49]. Thus, this gene may also be involved in the synthesis of fat and muscle protein, contributing to the structuring and body conformation of beef cattle.

The Chr14 harbors the most significant genomic region for CONF, with nine candidate genes. The *RAS-RAB2A* gene, a member of the RAS oncogene family, explained the highest proportion of the total additive genetic variance (3.89%) of CONF. Previous studies have reported its influence on beef quality due to its role in the synthesis of conjugated linoleic acid and in the positive regulation of muscle cell proliferation [50,51,52]. The cellular action of *RAB2A* was attributed to the biological function of protein transport from the endoplasmic reticulum to the Golgi apparatus to *RAB2* [53].

The importance of the *CYP7A1* gene for body structure was observed in GO analysis, with the gene being linked to pathways related to lipid metabolism, including cholesterol (GO:0006694) and lipid homeostasis (GO:0055088); steroid biosynthetic process (GO:0006694); monocarboxylic acid biosynthetic process (GO:0072330); and organic hydroxy compound biosynthetic process (GO:1901617). *CYP7A1* has been associated with the balance of thyroid hormones (thyroxine and triiodothyronine) in dairy cattle [54]. As a selection strategy, individuals carrying favorable alleles in QTLs located on Chr14 may have been more intensively selected for carcass and body conformation traits. This is also in line with the fact that the other genes found on Chr14 (*UBXN2B*, *TOX*, *FAM110B*, and *BTD1*) were previously reported to be associated with body weight, conformation, and carcass quality, including *TOX, FAM110B*, and *BTD1* which were associated with marbling [17,55,56]. Both *TOX* and *FAM110B* have also been linked to reproductive traits in cattle through their role in the transcription of the molecular regulation of puberty in Brahman cattle [57] and their influence on maternal ability and calving ease in Nellore females [58]. Six different pathways of cellular functions were assigned to the *UBXN2B* gene by GO analysis, highlighting its importance for the development and organization of vital cells. *KCNB2* (potassium voltage-gated channel subfamily B member 2) enables the transmembrane transport of potassium in muscle membranes by encoding protein A that makes up the ion channel [59].

The only candidate gene found on Chr20 was *CTNND2*, previously described as a gene belonging to a genomic region that is under constant selection because of its importance for growth and meat quality traits in Korean cattle [60]. Recently, it was observed the influence of this gene on the milk production of Holstein cows by its action on the concentrations of the thyroid hormone thyroxine [54]. The *MORF4L1* gene was found on Chr21, with the identification of multiple GO terms related to the acetylation of protein metabolites. Previous studies have associated *MORF4L1* with cattle birth weight because of its involvement in chromatin remodeling and cell regulation [61]. Another gene found on the same chromosome, *TM6SF1*, was previously associated with age at first calving in beef cattle [62]. *TM6SF1*, together with other candidate genes found in this study (*BTBD1* and *SH3GL3*), has also been associated with carcass and growth traits in broiler chickens [63]. Additionally, genes of the ADAMTS family (disintegrin and metalloproteinase with thrombospondin motifs) are widely expressed in mammalian tissues and can have multiple biological functions. There are reports of the influence of *ADAM* family genes on mechanisms related to fertilization, cell differentiation and adhesion, angiogenesis, immunity, and the development of epithelial and nervous tissues [64]. The gene detected in the present analysis was *ADAMTSL3,* which has variants considered as significant genetic marker of body traits in beef cattle [65]. In studies on growth traits in Brahman cattle, the same gene was associated with body weight measured at one year of age [66]. The effect of *ADAMTS12* on body structure has also been reported in pigs, and the authors demonstrated the influence of this gene on the differentiation and characterization of commercial and traditional breeds [67]. The effect of *ADAM* family genes was also reported on growth traits in beef cattle [68]. The authors described the role of the genes in this family on the regulation of adipogenesis through adipose tissue differentiation and skeletal muscle regeneration.

For PREC, ssGWAS enabled the identification of genes that are essential for growth and development, particularly those related to precocious growth. Four genes previously reported in the literature with an additive genetic effect on precocity were found on Chr2. The *PID1* gene has been identified in transgenic pigs, showing higher expression in obese animals [69], in which *PID1* overexpression resulted in the reduction in serum HDL-cholesterol and apolipoprotein A1 levels. The role of the *PID1* gene in human lipid metabolism is well known because this gene acts by reducing the sensitivity of adipocytes to insulin mediated by the interaction of domain 1 of phosphotyrosine with the lipoprotein receptor [70]. In GO analysis, *PID1* was identified in pathways associated with the intracellular transport of carbohydrates and regulation of metabolic processes of reactive oxygen species, confirming similar found in mice [71].

The *DNER* gene is involved in adipogenesis, mediating the differentiation of mesenchymal cells into adipocytes. In crossbred SimAngus cattle, the expression of this gene was inversely correlated with the degree of marbling, with the increase in the percentage of intramuscular fat being due to the multiplication of the number of differentiated adipocytes resulting from the decreased expression of *DNER* [72]. Inhibition of cell multiplication mediated by the action of *DNER* and the increase in adipogenesis were demonstrated by the upregulation of adipocyte markers, as well as by an increase in the frequency and size of adipocytes [73]. The *DNER* gene has also been associated with reproductive precocity in Japanese black cattle due to its influence on the regulation of age at first calving [74].

The expression of the *TRIP12* gene (thyroid hormone receptor interactor 12) was firstly identified in skeletal muscle of dairy and beef taurine breeds [75]. *TRIP12* is a gene involved in the regulation and differentiation of the musculature in mammals by ensuring the balance between protein synthesis and degradation [76]. The same theory was described in studies on effect of the *TRIP12* gene on the intramuscular fat content in Nellore cattle [58]. The same authors suggested that *TRIP12* influences the proteolysis of ubiquitin. In the genetic interaction network, *TRIP12* appears as the central gene, with direct connections with six other genes including *DNER*, demonstrating the importance of *TRIP12* in the interconnection between genes and their molecular actions. In view of the lack of information about the molecular role of *TRIP12* in the expression of visual score traits, further studies on the expression of this gene should be conducted. Furthermore, the effects of the *ACVR2A* gene have been reported on the reproductive development of cattle, with this gene being associated with testicular maturation [77] and with the regulation of granulosa cell proliferation [78].

On Chr3, the transcription coactivator *BCL9* was identified in pathways related to canonical cell signaling. Studies investigating muscle development in mice found that *BCL9* acts on myogenic modulation and differentiation during muscle development and regeneration [79]. *RNF115* has also been associated with intramuscular fat content in Nellore cattle [58]. *APBA1*, located on Chr8, has GO annotation in the cell–cell signaling pathway and has been associated with carcass traits (carcass weight and bone weight) in Simmental cattle [80].

The *PLCB4* gene, located on Chr13, has well-described functions in cattle. This gene has been directly associated with individual adaptive capacity to the environment, demonstrated by its influence on the physiological mechanisms involved in heat resistance [81,82] These findings highlight the effect of *PLCB4* on oxidative stress in cattle, suggesting a better and faster development of animals that are well adapted to climate adversities. Other important functions associated with *PLCB4* are linked to lipid metabolism, contributing to an increase in fat thickness and consequently in animal weight, with the amount of fat deposition increasing the added value of the product. The lack of expression of alterations in genes of the PLCB family is the main factor that explains the action of these lipase-forming genes on increasing muscle fat thickness [83].

The Chr23 harbors the genomic regions with the largest number of genes associated with PREC. The first gene identified on Chr23 was glutathione S-transferase alpha 5 (*GSTA5*), a gene previously linked to tenderness and other meat quality traits in French beef cattle breeds [84]. The *ELOVL5* gene was found in the region explaining the highest proportion of the total additive genetic variance for PREC. The genes of the elongation of very long chain fatty acid (*ELOVL*) family encode enzymes that play key roles in the elongation of very long chain fatty acids. The involvement of *ELOVL5* and *ELOVL6* in the synthesis of the main fatty acids present in beef as palmitic acid (C16:0), palmitoleic acid (C16:1), stearic acid (C18:0), and oleic acid, was reported in several cattle species [85]. The *ELOVL* gene family was significantly enriched in nine signaling pathways that regulate the synthesis, metabolism, and deposition of fat and has been previously associated with the lipid profile of Wagyu cattle [86]. The involvement of *ELOVL5* in different processes related to lipid metabolism also explains the association of the gene with milk fat content through its role in the regulation of triglycerides in epithelial cells of the mammary gland [87]. This gene also plays a decisive role in lipogenesis, where its expression tends to be potentiated during the postpartum period to improve the negative energy balance [88].

The *BoLA-DBQ* gene belongs to the bovine leukocyte antigen (*BoLA*) family and is characterized by a complex structure of linkage disequilibrium and a high degree of genetic variation [89]. This gene located on Chr23 was largely associated with physiological and immunological mechanisms of resistance against diseases that have a significant impact on animal production [90,91]. Although *BoLA-DBQ* was not described in the GO analysis of the present study, different pathways involved in cellular processes, protein synthesis, and immune responses to external pathogens have been associated with other genes of the *BoLA* family [92]. The last genes associated with PREC and already described in the literature were *MDGA1*, *ZFAND3*, and *BTBD9*, with reported effects on reproductive traits and weight gain. The *MDGA1* have been shown to exert significant effects on backfat thickness in native Korean cattle [93]. The *ZFAND3* gene has been associated with growth traits in multibreed pigs, especially those related to weight gain and feed efficiency [94]. The BTB domain, present in the *BTBD9* gene, is associated with synaptic plasticity [95], as well as with embryo lethality in different cattle breeds [96]. *BTBD9* has also been shown to have proteolytic properties, which affect the organoleptic characteristics of beef such as tenderness and water-holding capacity [97]. GO annotation revealed two cell signaling GO terms for *BTBD9*: canonical Wnt signaling (GO:0060070) and cell-cell signaling (GO:0007267).

For MUSC, the neuromodulin gene (*GAP43*) was found to be associated with neuronal growth. This important protein, present on the inner surface of the plasma membrane of axon terminals, is involved in axon orientation, neuroregeneration, and control of neurotransmitter release. For a long time, *GAP43* has been considered neuron specific; however, recent studies suggest a role of the gene in interactions with lipid groups and identified its expression in satellite cells and skeletal muscle cells present on muscle fibers of mice [98,99]. Furthermore, an influence of this gene on cattle behavior was previously reported, indicating its importance not only in the muscle development as in the neurological pattern [100].

Like the *GAP43* gene that exerts known GO functions related to the synthesis and development of the nervous system, GO analysis identified three other candidate genes (*SEMA6A*, *TIAM2*, *UNC5A*) linked to GO terms involved in the following nervous pathways: generation of neurons (GO:0048699), neurogenesis (GO:0022008), and nervous system development (GO:0007399). This fact supports the hypothesis that *GAP43* not only exerts neuronal functions but is also involved in pathways linked to the development of the anatomical structure.

The *LSAMP* gene (limbic system-associated membrane protein) is expressed by cortical and subcortical neurons in areas linked to memory, cognitive behavior, learning, central autonomous regulation, and formation of neurosensory connections [101,102]. *LSAMP* has been associated with the response to heat stress in male Duroc pigs [17] and with the composition of fatty acids in Tibetan pigs due to its involvement in the synthesis of octadecatrienoic acid (18:3) [103].

On Chr7, the *SEMA6* gene encodes semaphorin 6A, expressed in the intermediate zone of the cerebral cortex [104], where it acts on the differentiation and myelination of oligodendrocytes. *SEMA6* has been associated with height in horses and with feed efficiency in pigs, where its regulation in skeletal muscle was low [105]. The presence of the *SEMA6A* gene in different GO terms was associated with various nervous system functions. This candidate gene harbors the genomic window explaining the highest percentage of the total additive genetic variance for MUSC, involved in all physiological and metabolic processes of the ontological coverage in related pathways, such as RNA processing, response to external stimulus, cellular component movement, generation of precursor metabolites and energy, cyclic organic compound metabolic process, and body growth and developmental processes.

The *FAF2* gene has been linked to milk composition in goats through its action on the oxidative phosphorylation pathway [106]. In view of the lack of reports associating *FAF2* with muscle growth and/or development, an in-depth study is necessary to investigate the potential relationship of this gene with cattle body development. Another gene present on the same chromosome, *HIGD2* encodes hypoxia inducible mitochondrial protein. In both pigs and cattle, the involvement of this gene in the assembly of the respiratory complex was confirmed by demonstrating its influence on swine pH and the adaptive capacity of African cattle in responding to heat [107,108]. The *HIGD2A* is induced under stress conditions such as glucose deprivation or hypoxia in epithelial cells when *HIGD2A* is upregulated, promoting cell survival [109].

One gene in GO terms related to pathways of neurological processes is *UNC5A* (Unc-5 netrin receptor A), which guides axonal navigation during neural development. In the present study, this gene was linked to 29 GO terms. The same gene was highlighted in the literature for its importance in the feed efficiency of pigs [110] and muscle development in subpopulations of Brazilian and Canadian Angus cattle [111], indicating a relationship of *UNC5A* with growth performance.

The last two genes identified on Chr7, *HK3*, and *UIMC1*, belong to the family of hexokinases and ubiquitin interaction motif, respectively. Both genes have been associated with finishing in Hanwoo cattle [16]. The role of hexokinases in the phosphorylation of glucose inside the cell for metabolic use has been earlier described in the literature [112,113]. The hexokinases present essential functions to these proteins, catalyzing the phosphorylation of hexose that results in sucrose-induced signal transduction when regulated by the expression of the *HK3* gene. Studying the expression of hypoxia-inducible genes in cattle, the *HK2* gene was found to be essential for adaptation of the animal to high altitudes because of the role of the gene in maintaining cellular homeostasis in cattle [114].

Regarding *UIMC1*, there is only one additional report of an association with sperm quality and motility in boars [115]. The importance of this gene was revealed by GO analysis, which linked it to 15 GO terms: histone modification (GO:0016570), covalent chromatin modification (GO:0016569), cellular response to stress (GO:0033554), response to stimulus (GO:0050896), double-strand break repair (GO:0006302), regulation of DNA metabolic process (GO:0051052), deubiquitination (GO:0016579), and protein modification (GO:0070646). Thus, despite the lack of studies on the action of this gene in livestock species, the considerable number of GO annotations mainly linked to the regulation of metabolism and protein modification may render *UIMC1* a possible candidate gene associated with muscle development in beef cattle.

On Chr9, the *TIAM2* gene was linked to multiple GO terms related to nervous system development and cell morphogenesis. *TIAM2* functionally was associated with lipid metabolism in different species [116]. Additionally, according to the GO analysis, the *SCAF8* gene plays a role in messenger RNA processing. In the genetic interaction network, *SCAF8* showed a strong relationship with the *TIAM2* gene detected on the same chromosome. Both genes were also previously indicated as responsibles for lean muscle content, which corroborates to the hypothesis that *TIAM2* connected with *SCAF8* can represent a important role on tissue constitution [117].

The last chromosome harboring SNPs associated with muscling was Chr21, where the *STXBP6* gene has already been linked to marbling quality in a study on selection signatures in Korean cattle [118]. Also located on Chr21, *STOML1* (stomatin-like protein 1) encodes a stomatin-related membrane protein whose function is the modulation of acid-sensing ion channels [119,120]. When binding to another gene called *TEAD1*, *STOML1* acts as a key molecule in cell proliferation and differentiation that result in muscle development [121]. The *LOXL1* is fundamental for the crosslinking between collagen and elastin, ensuring the elasticity of the extracellular matrix in organs where the gene is expressed [122,123]. The influence of *LOXL1* was described on copper concentrations in the liver and pulmonary artery of swine through the establishment of elastic fibers, which may be related to the adipogenesis of subcutaneous preadipocytes linked to ontogenetic pathways [124].

The inclusion of genomic information in animal breeding programs is a wise strategy to improve the selection of economic important traits [125]. In this context, the identification of genomic regions, as well as of the biological processes that affect conformation, precocity, and muscling, is essential to facilitate the understanding of the biological mechanisms underlying the expression of these traits. The present study highlighted genomic regions that harbor genes previously reported in the literature, as well as some new candidate genes that may help elucidate the genetic knowledge of visual score traits, such as those observed here in the Nellore breed. The variants in candidate genes identified in this study can be used in the future in genetic breeding programs for improving carcass- and growth-related traits in beef cattle. In addition, because of mutations in the candidate genes may be affecting some process related to CONF, PREC, and MUSC validation studies through gene expression analyzes must be carried out in a group of animals. These validations are essential to confirm the participation of new candidate genes in the development of the evaluated traits.

## 5. Conclusions

The visual scores of conformation, precocity, and muscling can be genetically improved through direct genetic selection since their heritabilities are moderate. The ssGWAS identified 27 genomic regions for conformation, highlighting *ALDH9A1*, *RXRG*, *RAB2A*, and *CYP7A1* as candidate genes due to their relevance in biological processes. The main candidate gene for precocity was *ELOVL5* due to its role in the synthesis of long-chain fatty acids, a fundamental process in the establishment of the lipid profile of beef. In addition to *ELOVL5*, four other candidate genes were identified, including *PID1*, *DNER*, *TRIP12*, and *PLCB4*. For muscling, the *SEMA6A*, *TIAM2*, *UNC5A,* and *UIMC1* genes were the most involved in different biological pathways that result in the expression and development of muscles in cattle. Taken together, the genomic regions identified are of great importance for the understanding of the molecular mechanisms underlying visual score traits in Nellore cattle and may be useful to improve the genomic predictions for traits of economic interest in beef cattle.

## Figures and Tables

**Figure 1 animals-12-03526-f001:**
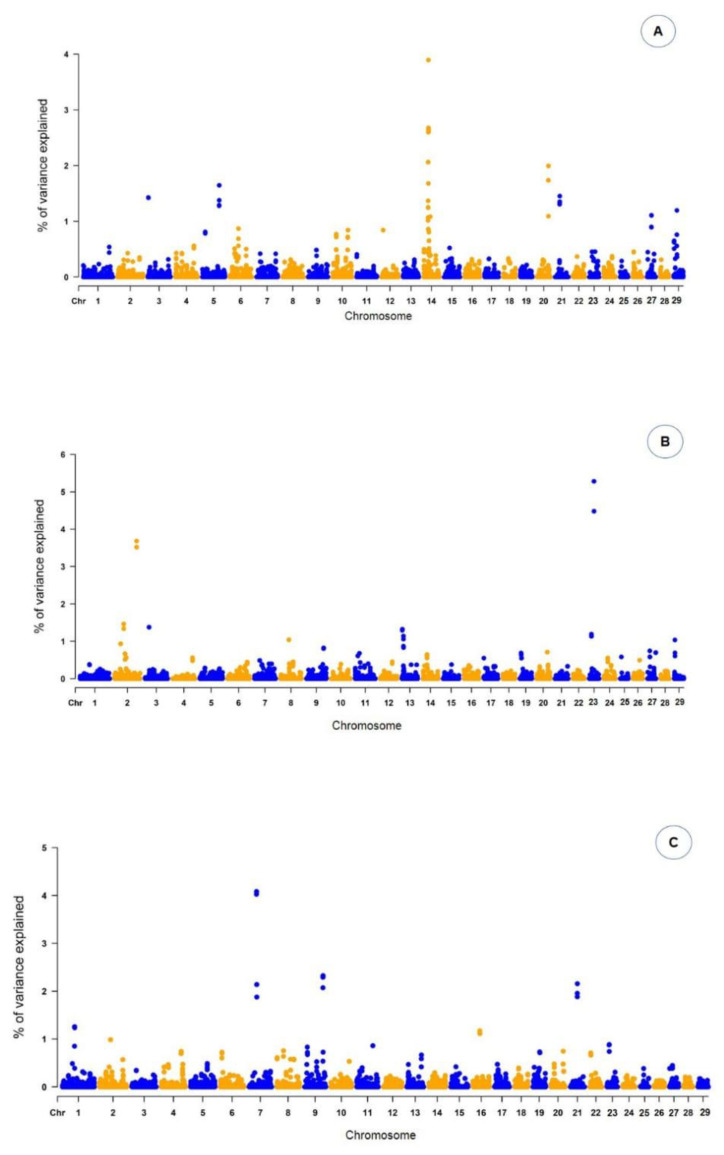
Manhattan plot of the percentage of the total additive genetic variance explained by windows of 10 adjacent SNPs for conformation (**A**), precocity (**B**) and muscling (**C**) scores in Nellore cattle. *X*-axis: autosomes. *Y*-axis: percentage of additive genetic variance explained.

**Figure 2 animals-12-03526-f002:**
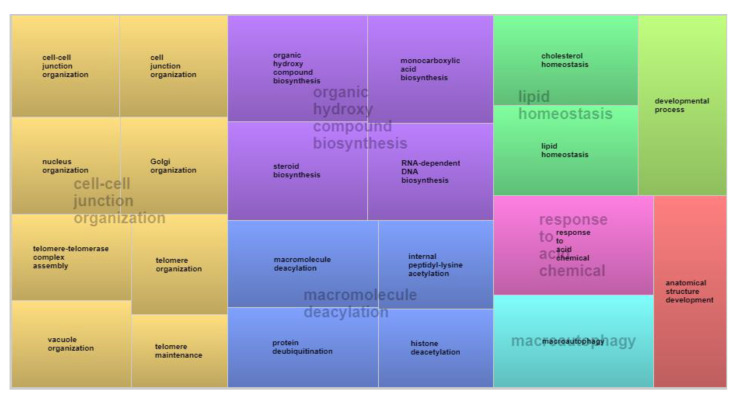
Analysis of biological processes co-associated with body conformation evaluated based on visual scores in Nellore cattle.

**Figure 3 animals-12-03526-f003:**
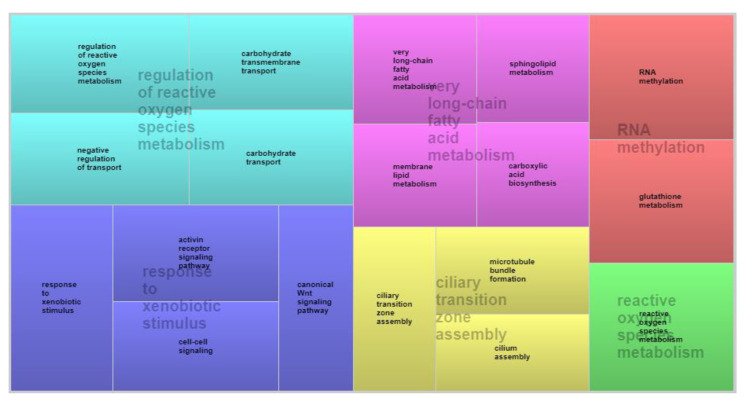
Analysis of biological processes co-associated with precocity evaluated based on visual scores in Nellore cattle.

**Figure 4 animals-12-03526-f004:**
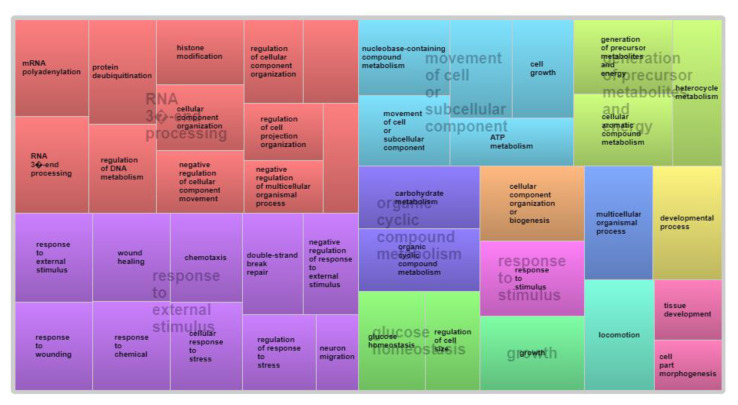
Analysis of biological processes co-associated with muscling evaluated based on visual scores in Nellore cattle.

**Figure 5 animals-12-03526-f005:**
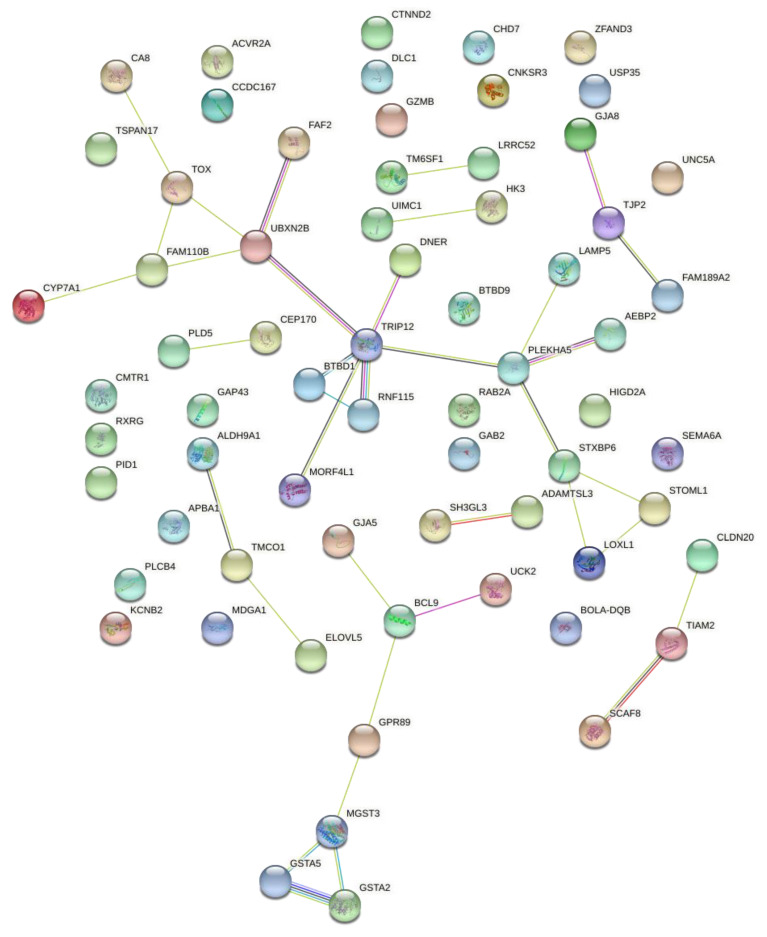
Genetic interaction network of genes simultaneously associated with conformation, precocity, and muscling traits based on visual scores in Nellore cattle.

**Table 1 animals-12-03526-t001:** Descriptive statistics, variance components and heritabilities for visual scores of conformation (CONF), precocity (PREC), and muscling (MUSC) in Nellore cattle.

Variable	N	Mean	±SD	Minimum	Maximum	σ^2^_a_	σ^2^_e_	σ^2^_p_	h^2^ (S.E.)
CONF (score)	20,808	3.39	0.96	1	5	0.28	0.56	0.84	0.33 + 0.01
PREC (score)	20,808	3.46	0.97	1	5	0.32	0.53	0.85	0.37 + 0.01
MUSC (score)	20,808	3.11	0.98	1	5	0.33	0.52	0.85	0.38 + 0.02

N—number of animals in the analysis; SD—standard deviation; σ^2^_a_—additive genetic variance, σ^2^_e_—residual variance; σ^2^_p_—phenotypic variance; h^2^—heritability; S.E.—standard error.

**Table 2 animals-12-03526-t002:** Identification and description of genes located in windows associated with visual scores for conformation (CONF) that explained more than 1% of the additive genetic variance.

Chr	SNP Positions	Var (%)	Gene	Gene Name
3	3,033,833:3,060,955	1.427	*UCK2*	Uridine-cytidine-kinase 2
3	3,164,474:3,192,425	1.427	*TMCO1*	Transmembrane and coiled-coil domains 1 *Bos taurus* (TMCO1), mRNA
3	3,212,293:3,249,677	1.427	*ALDH9A1*	Aldehyde dehydrogenase 9 family member A1
3	3,310,029:3,310,881	1.427	*MGST3*	Microsomal glutathione S-transferase
3	3,431,614:3,345,407	1.427	*LRRC52*	Leucine-rich repeat containing 52
3	3,470,729:3,536,373	1.423	*RXRG*	Retinoid X receptor gamma
5	90,271,919:90,351,449	1.291	*AEBP2*	AE binding protein 2
5	90,610,566:90,837,005	1.646	*PLEKHA5*	Pleckstrin homology domain-containing family A member 5
14	23,883,121:24,011,986	1.016	*BPNT2*	3′(2′), 5′-bisphosphate nucleotidase 2
14	24,373,031:24,396,836	1.251	*FAM110B*	Family with sequence similarity 110 member B
14	24,590,812:24,624,435	1.368	*UBXN2B*	UBX domain-containing protein 2B
14	24,651,537:24,675,169	2.062	*CYP7A1*	Cholesterol 7α-hydroxylase
14	25,079,291:25,258,596	1.245	*TOX*	Thymocyte selection-associated high mobility group box
14	25,866,853:25,887,784	2.601	*CA8*	Carbonic anhydrase 8
14	26,217,826:26,253,265	3.898	*RAB2A*	Member RAS oncogene family
14	26,453,389:26,482,710	2.677	*CHD7*	Chromodomain-helicase DNA binding protein 7
14	36,011,003:36,195,954	1.083	*KCNB2*	Potassium voltage-gated channel subfamily B member 2
20	61,641,222:61,706,531	1.995	*CTNND2*	Catenin delta 2
21	24,484,472:24,509,041	1.328	*ADAMTSL3*	ADAMTS-like 3
21	24,597,677:24,614,322	1.307	*SH3GL3*	SH3 domain containing GRB2-like 3, endophilin A3
21	24,843,672:24,921,366	1.307	*HDGFL3*	HDGF-like 3
21	24,933,056:24,947,938	1.328	*TM6SF1*	Transmembrane 6 superfamily member 1
21	25,045,675:25,045,726	1.452	*BTBD1*	Pleckstrin homology domain containing A5
21	25,194,048:25,229,486	1.352	*MORF4L1*	Mortality factor 4-like 1
27	23,270,756:23,916,949	1.107	*DLC1*	DLC1 Rho GTPase activating protein
29	17,689,798:17,792,365	1.197	*GAB2*	GRB2-associated binding protein 2
29	17,825,902:17,853,459	1.197	*USP35*	Ubiquitin-specific peptidase 35

**Table 3 animals-12-03526-t003:** Identification and description of genes located in windows associated with visual scores for precocity (PREC) that explained more than 1% of the additive genetic variance.

Chr	SNP Positions	Var (%)	Gene	Gene Name
2	48,421,237:48,808,281	1.465	*ACVR2A*	Activin A receptor type 2A
2	117,076,082:117,094,178	3.683	*PID1*	Phosphotyrosine interaction domain containing 1
2	117,441,080:117,623,992	3.521	*DNER*	Delta/notch-like EGF repeat containing
2	117,857,150:117,859,391	3.521	*TRIP12*	Thyroid hormone receptor interactor 12
3	21,542,305:21,628,711	1.377	*RNF115*	Ring finger protein 115
3	21,653,070:21,722,429	1.377	*GPR89A*	G-protein coupled receptor 89A
3	21,778,330:21,791,863	1.377	*GJA8*	Gap junction protein alpha 8
3	21,859,096:21,925,914	1.377	*GJA5*	Gap junction protein alpha 5
3	22,006,767:22,098,901	1.379	*BCL9*	BCL9 transcription coactivator
8	45,324,486:45,403,166	1.040	*TJP2*	Tight junction protein 2
8	45,494,923:45,527,302	1.041	*FAM189A2*	Family with sequence similarity 189 member A2
8	45,648,545:45,806,337	1.041	*APBA1*	Amyloid beta precursor protein binding, family A, member 1
13	1,912,749:2,287,678	1.310	*PLCB4*	Phospholipase C beta 4
13	2,522,361:2,553,000	1.324	*LAMP5*	Lysosome-associated membrane protein family member 5
13	7,845,572:7,852,824	1.142	*FRLT3*	Fibronectin-leucine-rich transmembrane protein 3
23	11,334,019:11,372,125	1.192	*CMTR1*	Cap methyltransferase 1
23	11,389,074:11,436,996	1.192	*CCDC167*	Coiled-coil domain containing 167
23	11,560,373:11,597,202	1.192	*MDGA1*	MAM domain containing glycosylphosphatidylinositol anchor 1
23	11,735,041:12,026,180	1.132	*ZFAND3*	AN1-type zinc finger protein 3
23	12,119,175:12,435,186	1.132	*BTBD9*	BTB domain containing 9
23	25,088,616:25,099,203	5.281	*GSTA2*	Glutathione S-transferase alpha 2
23	25,165,091:25,171,927	5.282	*GSTA5*	Glutathione S-transferase alpha 5
23	25,232,601:25,272,669	5.281	*CILK1*	Ciliogenesis-associated kinase 1
23	25,438,003:25,474,020	5.281	*ELOVL5*	Elongation of very long chain fatty acids 5
23	25,541,686:25,669,376	5.281	*BOLA-DQB*	Major histocompatibility complex, Class II, DQ beta

**Table 4 animals-12-03526-t004:** Identification and description of genes located in windows associated with visual scores for muscularity (MUSC) that explained more than 1% of the additive genetic variance.

Chr	SNP Positions	Var (%)	Gene	Gene Name
1	60,273,018:60,301,377	1.259	*GAP43*	Growth-associated protein 43
1	60,476,799:60,504,874	1.236	*LSAMP*	Limbic system-associated membrane protein
7	36,887,774:37,185,124	4.083	*SEMA6A*	Semaphorin 6A
7	37,857,202:37,909,672	2.139	*HIGD2A*	HIG1 hypoxia-inducible domain family member 2a
7	37,928,274:37,983,978	2.139	*FAF2*	Fas-associated factor 2
7	38,116,057:38,126,424	2.138	*TSPAN17*	Tetraspanin 17
7	38,306,218:38,331,961	1.879	*UNC5A*	Unc-5 netrin receptor A
7	38,345,801:38,352,118	1.878	*HK3*	Hexokinase 3
7	38,385,427:38,410,913	1.878	*UIMC1*	Ubiquitin interaction motif containing 1
9	91,134,263:91,182,225	2.292	*CNKSR3*	CNKSR 3 family member
9	91,496,960:91,538,184	2.311	*SCAF8*	SR-related CTD-associated factor 8
9	91,663,611:91,732,039	2.324	*TIAM2*	TIAM Rac1 associated GEF 2
9	91,886,978:91,936,136	2.323	*CLDN20*	Claudin 20
16	34,020,390:34,166,299	1.170	*CEP170*	Centrosomal protein 170
16	34,431,274:34,439,744	1.114	*PLD5*	Phospholipase D family member 5
21	34,641,526:34,712,491	1.957	*STOML1*	Stomatin-like 1
21	34,664,967:34,672,830	1.884	*LOXL1*	Lysyl oxidase-like 1
21	34,785,250:34,813,843	1.956	*GZMB*	Granzyme B
21	35,032,718:35,183,150	2.157	*STXBP6*	Syntax-binding protein 6

## Data Availability

The genotype data are available in the OSF Repository (https://osf.io/b8z9y/), accessed on 24 May 2022.

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
