# Peer review of "Genome-Wide Association Analysis Reveals Novel Loci Related with Visual Score Traits in Nellore Cattle Raised in Pasture–Based Systems"

_animals, 2022, doi:10.3390/ani12243526_

Round 1

Reviewer 1 Report

The present study was to perform a single-step GWAS (ssGWAS) to identify genomic regions, candidate genes, and their biological functions associated with conformation, precocity, and muscling scores in Nellore cattle raised in pasture-based systems. The sample size of the manuscript is large, the analysis results are more thorough, and the conclusion is reliable. I did a careful search of the database(https://www.ncbi.nlm.nih.gov/home/about/, ect) and found no similar studies to gain. I recommend the manuscript for publication.

1. Please shorten the conclusion, it's too long.

2. Could you quote the following article?

Sara de las Heras-Saldana, Samuel A. Clark, Naomi Duijvesteijn, Cedric Gondro, Julius H. J. van der Werf, Yizhou Chen.Combining information from genome-wide association and multi-tissue gene expression studies to elucidate factors underlying genetic variation for residual feed intake in Australian Angus cattle.BMC Genomics. 2019; 20: 939. Published online 2019 Dec 6. doi: 10.1186/s12864-019-6270-4

Author Response

The present study was to perform a single-step GWAS (ssGWAS) to identify genomic regions, candidate genes, and their biological functions associated with conformation, precocity, and muscling scores in Nellore cattle raised in pasture-based systems. The sample size of the manuscript is large, the analysis results are more thorough, and the conclusion is reliable. I did a careful search of the database (https://www.ncbi.nlm.nih.gov/home/about/, ect) and found no similar studies to gain. I recommend the manuscript for publication.

Authors: Dear reviewer, we appreciate your time and suggestions. All recommendations were attended as described below.

1. Please shorten the conclusion, it's too long.

Authors: The conclusion was reduced as requested.

2. Could you quote the following article?

Sara de las Heras-Saldana, Samuel A. Clark, Naomi Duijvesteijn, Cedric Gondro, Julius H. J. van der Werf, Yizhou Chen.Combining information from genome-wide association and multi-tissue gene expression studies to elucidate factors underlying genetic variation for residual feed intake in Australian Angus cattle.BMC Genomics. 2019; 20: 939. Published online 2019 Dec 6. doi: 10.1186/s12864-019-6270-4

Authors: The manuscript was quoted as requested.

Reviewer 2 Report

Machado et al. performed GWAS for the Visual Score Traits in cattle and identified several SNPs and genes related to these traits. Overall, the manuscript is well written, and the selection of methods (single-step GWAS and PANTHER + REVIGO) is appropriate. The authors also well discussed these results. I have a few minor comments.

Line 27: single nucleotide polymorphisms (SNP):  the SNP abbreviation in the abstract is not necessary.

Line 20-25: Might add the version of the software as well.

Line 68: Several markers: I think it is quite a lot, might be many is a better word.

Line 70: The authors might provide an update on the number of QTLs or SNPs that have been identified for these traits in the QTL database.  

One important point from this study is the trait is derived from the visual scores, the authors might introduce (a few sentences) the application of visual scores and why it is important and can be used in breeding programs.

Line 193-194: If this extent of LD is the result of other studies, the authors might provide references for it.

Line 198-199: Did the authors consider all levels of GO terms in enrichment analyses? For REVIGO, which parameters did the authors use?

Line 200: Missing references for String

Line 322: Might change Indigenous to indigenous

Author Response

Machado et al. performed GWAS for the Visual Score Traits in cattle and identified several SNPs and genes related to these traits. Overall, the manuscript is well written, and the selection of methods (single-step GWAS and PANTHER + REVIGO) is appropriate. The authors also well discussed these results. I have a few minor comments.

Authors: Dear reviewer, we appreciate your efforts to indicate crucial points that should be improved in the manuscript. All suggestions were attended as described in detail below.

Line 27: single nucleotide polymorphisms (SNP):  the SNP abbreviation in the abstract is not necessary.

Authors: We modified as requested.

Line 20-25: Might add the version of the software as well.

Authors: The information was added.

Line 68: Several markers: I think it is quite a lot, might be many is a better word.

Authors: Thank you. The text was modified as suggested.

Line 70: The authors might provide an update on the number of QTLs or SNPs that have been identified for these traits in the QTL database.  

Authors: The number of QTLs related with meat and carcass was included in the text as suggested.

One important point from this study is the trait is derived from the visual scores, the authors might introduce (a few sentences) the application of visual scores and why it is important and can be used in breeding programs.

Authors: Thank you. The information was included in the main text.

Line 193-194: If this extent of LD is the result of other studies, the authors might provide references for it.

Authors: The threshold was defined based on the extent of linkage disequilibrium from the studied population as stated in the text.

Line 198-199: Did the authors consider all levels of GO terms in enrichment analyses? For REVIGO, which parameters did the authors use?

Authors: All levels of GO terms were considered and for REVIGO, the default parameters with uncorrected p-values were utilized.

Line 200: Missing references for String

Authors: The reference was included as suggested.

Line 322: Might change Indigenous to indigenous

Authors: We have modified as suggested.

Reviewer 3 Report

Here are some details that the authors must clarify, when a gwas analysis is done, generally, the mutations associated with a characteristic are identified, and specific methodologies are applied to a group of animals to select which ones comply or which do not, and so it is done groups, with or without characteristics. It strikes me that they do not explain how they selected the animals for each characteristic (conformation, precocity and muscularity). I have these observations to improve

Line 74: What is meant by conformation, precocity and muscularity, and why these characteristics are important.

Materials and Methods: 

It remains to describe how they evaluated the characteristics of conformation, precosity and musculity in the 20,807 animals. What were the reference values for each parameter?, que valor which allowed them to be considered in the study groups. Which were the control group and the case study. Also, for each evaluation of each parameter, were females and males considered in the same group? can mixing males and females influence the GWAS analysis? please explain.

It is understood that in gwas analysis, it is to identify gene mutations that are associated with a trait such as conformation, precocity, and muscalarity. What role does the diet play? 35% Brachiaria brizantha and 65% Panicum maximum, is it a factor that induces these mutations? If you feed other types of food, are you sure that you will identify the same mutations in the genes that you report? explain more, to understand the study

Line 225: To avoid confusion, it is recommended to change BTA to Chr, chromosome.

Line 232: It is recommended to complete the information in tables 2, 3 and 4, with the most significant SNPs (top ten gene), the position in the genome, A1, minor allele 1; A2, major allele 2.

Line 246: It is understood that they only analyzed the data, however, how did they validate at the laboratory level that the genes reported in tables 2 to 4 do indeed have a mutation associated with each parameter? Please, how can you validate it or why did you not validate it? , an idea for example when analyzing gene expression with rnaseq, is validated with a qpcr. But when DNA is analyzed, how is it validated?

In table 4, he myostatin gene would be expected to appear, commonly associated with mutations in muscle proteins.

Line 573: candidate genes (ALDH9A1, RXRG, RAB2A, and CYP7A1) for which trait? define

Line 576: Candidate genes (ELOVL5 PID1, DNER, TRIP12, and PLCB4) for precocity? It strikes me that if these are genes associated with sexual precocity, there are none related to hormonal processes in the formation of sexual gametes and reproductive processes. If you put males and females in the same group, how can you define that these genes are large amounts of precociousness, if each sex has different mechanisms? please explain this.

Author Response

Here are some details that the authors must clarify, when a gwas analysis is done, generally, the mutations associated with a characteristic are identified, and specific methodologies are applied to a group of animals to select which ones comply or which do not, and so it is done groups, with or without characteristics. It strikes me that they do not explain how they selected the animals for each characteristic (conformation, precocity and muscularity). I have these observations to improve.

Authors: Dear reviewer, we appreciate your comments and suggestions. We understand the reviewer’s concern about gene validation. However, a large number of manuscripts have used the same approach utilized in our research, due to the great importance in detecting genomic regions that may be associated with traits of economic interest. In addition, these studies can better address the genes that must be further validated, facilitating and improving the entire process of knowledge of the molecular structures involved in the formation of relevant livestock traits. We would like to highlight that more information about the importance of validating these regions has been included in the text as suggested by the reviewer.

Line 74: What is meant by conformation, precocity and muscularity, and why these characteristics are important.

Authors: A better definition of each trait was included in the Material and Methods section and the importance of these traits was highlighted in the introduction.

Materials and Methods: 

It remains to describe how they evaluated the characteristics of conformation, precosity and musculity in the 20,807 animals. What were the reference values for each parameter?, que valor which allowed them to be considered in the study groups. Which were the control group and the case study. Also, for each evaluation of each parameter, were females and males considered in the same group? can mixing males and females influence the GWAS analysis? please explain.

Authors: We agreed that a better definition of the traits was needed, as well as the groups in which these animals were evaluated. As mentioned above, this information was included in the M&M section. Thank you.

It is understood that in gwas analysis, it is to identify gene mutations that are associated with a trait such as conformation, precocity, and muscalarity. What role does the diet play? 35% Brachiaria brizantha and 65% Panicum maximum, is it a factor that induces these mutations? If you feed other types of food, are you sure that you will identify the same mutations in the genes that you report? explain more, to understand the study

Authors: The animal feeding information brings a context of the system in which these animals were raised. Information on the impact of the nutritional system on the GWAS analyzes has been included in the Discussion section.

Line 225: To avoid confusion, it is recommended to change BTA to Chr, chromosome.

Authors: The modifications were made as recommended.

Line 232: It is recommended to complete the information in tables 2, 3 and 4, with the most significant SNPs (top ten gene), the position in the genome, A1, minor allele 1; A2, major allele 2.

Authors: The most significant SNPs are represented by the % of additive genetic variance explained, in which the higher the % the higher the significance. The position is represented in the “genomic region” column. Unfortunately, we did not have access to the minor and major alleles information. The database provided to us only contained the map (SNP-name, Chr, and BP position) and the transformed genomic file (0,1,2,and 5).

Line 246: It is understood that they only analyzed the data, however, how did they validate at the laboratory level that the genes reported in tables 2 to 4 do indeed have a mutation associated with each parameter? Please, how can you validate it or why did you not validate it? , an idea for example when analyzing gene expression with rnaseq, is validated with a qpcr. But when DNA is analyzed, how is it validated?

Authors: The importance of validating candidate genes through gene expression was included in the text. We agree with the reviewer that in addition to identifying genomic regions associated with traits of economic importance through GWAS, validation studies should be performed to confirm the effectiveness of the genes for the evaluated traits. We are performing laboratory analyzes to confirm and validate the genes found here. The results of this additional research are expected to be published in the near future.

In table 4, he myostatin gene would be expected to appear, commonly associated with mutations in muscle proteins.

Authors: We agree that the myostatin gene appears frequently in studies that evaluate the bovine meat. However, as our study does not directly evaluate the muscle, but rather muscle development as a growth characteristic, this may explain the absence of the gene in the results obtained here.

Line 573: candidate genes (ALDH9A1, RXRG, RAB2A, and CYP7A1) for which trait? Define

Authors: The trait was described in the text: “genomic regions for conformation, highlighting ALDH9A1, RXRG, RAB2A, and CYP7A1”.

Line 576: Candidate genes (ELOVL5 PID1, DNER, TRIP12, and PLCB4) for precocity? It strikes me that if these are genes associated with sexual precocity, there are none related to hormonal processes in the formation of sexual gametes and reproductive processes. If you put males and females in the same group, how can you define that these genes are large amounts of precociousness, if each sex has different mechanisms? please explain this.

Authors: The precocity trait is not related to sexual precocity but with finishing precocity (which is related to fat deposition). This information was added in the beginning of the manuscript to avoid misunderstanding.

Round 2

Reviewer 3 Report

In materials and methods: It is not yet clear how they organized the experimental group of animals. compared groups 1 (lower) vs 5 (higher)? How many animals for each group or not analyzed in a pool? Generally, in GWAS studies, two groups are always compared, presence or absence, low or high, in relation to the characteristic to be evaluated. please provide more detail how they did it.

The tables still lack the column for the minor allele and the major allele. It is important to mention the position of the SNPs, within each gene.

Line 596. In conclusions. It is important to clarify that some mutation in the candidate genes may be affecting some process related to precocity, conformation and musculatin. However, it is important to do a validation test for the presence of these mutations in these genes in a group of animals.

Author Response

Authors: Dear reviewer, we appreciate your efforts to indicate crucial points that should be improved in the manuscript. We did our best to meet your expectations. Please, find details in below.

In materials and methods: It is not yet clear how they organized the experimental group of animals. compared groups 1 (lower) vs 5 (higher)? How many animals for each group or not analyzed in a pool? Generally, in GWAS studies, two groups are always compared, presence or absence, low or high, in relation to the characteristic to be evaluated. please provide more detail how they did it.

Authors: In this study we did not perform selective genotyping nor DNA pooling. The genotyped animals had all possible phenotypic records, as commonly done in livestock GWAS studies.

The tables still lack the column for the minor allele and the major allele. It is important to mention the position of the SNPs, within each gene.

Authors: The SNP position was included in the tables as requested. Unfortunately, as mentioned before, we did not have access to the minor and major alleles information. The database available to us only contained the map information only (SNP-name, Chr, and BP position) and the recoded genomic information (0,1,2,and 5). Nevertheless, we believe that even in the absence of this information, the search for genes and the functional analyzes are not compromised, which does not reduce the quality of the results. Thank you in advance for your understanding.

Line 596. In conclusions. It is important to clarify that some mutation in the candidate genes may be affecting some process related to precocity, conformation and musculatin. However, it is important to do a validation test for the presence of these mutations in these genes in a group of animals.

Authors: The additional information was added to the text. We appreciate your suggestions.